# Norovirus Is the Most Frequent Cause of Diarrhea in Hospitalized Patients in Monterrey, Mexico

**DOI:** 10.3390/pathogens9090672

**Published:** 2020-08-19

**Authors:** Néstor Casillas-Vega, Fernanda Flores-Rodríguez, Israel Sotelo-Coronado, Magda Elizabeth Vera-García, Aldo García-Heredia, Ana Ma. Rivas-Estilla, Sonia A. Lozano-Sepúlveda, Santos García, Amador Flores-Arechiga, Norma Heredia

**Affiliations:** 1Departamento de Patología Clínica, Hospital Universitario Dr. José Eleuterio González, Universidad Autónoma de Nuevo León, 64460 Monterrey, Nuevo León, Mexico; nestor.casillasvg@uanl.edu.mx (N.C.-V.); jesus.sotelocr@uanl.edu.mx (I.S.-C.); algaredia@gmail.com (A.G.-H.); arechiga@outlook.com (A.F.-A.); 2Departamento de Microbiología e Inmunología, Facultad de Ciencias Biológicas, Universidad Autónoma de Nuevo León, 66450 San Nicolás, Nuevo León, Mexico; mflorerodriguez@gmail.com (F.F.-R.); santos@microbiosymas.com (S.G.); 3Departamento de Bioquímica y Medicina Molecular, Facultad de Medicina, Universidad Autónoma de Nuevo León, 64460 Monterrey, Nuevo León, Mexico; magda.veragrc@uanl.edu.mx (M.E.V.-G.); ana.rivasst@uanl.edu.mx (A.M.R.-E.); sonia.lozanosp@uanl.edu.mx (S.A.L.-S.)

**Keywords:** pathogens, diarrhea, gastrointestinal infection, Mexico

## Abstract

Little information is available regarding the pathogens that cause diarrhea in hospitalized patients who also have various clinical problems. The purpose of this study was to determine the presence of pathogens in fecal samples of hospitalized patients all suffering diarrhea in addition to other problems in Mexico. Diarrheic stools from 240 patients were obtained in a third-level hospital in Monterrey, Mexico. PCR was used for the detection of *Salmonella* spp., *Shigella* spp., *Campylobacter* spp., *Yersinia* spp., *Aeromonas* spp., *Clostridioides difficile*, and norovirus GI and GII. The presence of trophozoites, cysts of protozoa, eggs, and/or helminth larvae was determined by microscopic observation. Of the 240 patients analyzed, 40.4% presented at least one of the pathogens analyzed. Norovirus was the pathogen most frequently found (28.6%), followed by bacteria (11.7%), and parasites (8.3%). The majority of co-infections were parasites + norovirus, and bacteria + norovirus. Norovirus was detected mainly in children aged 0 to 10 years (9/15, 60%). Patients aged 0–20 years did not present co-infections. *Entamoeba coli* and *Entamoeba histolytica* were the most common parasites, (8/240), and *Salmonella* was the most prevalent bacteria (10/240). This information can help design specific strategies useful for hospitalized people with a compromised status.

## 1. Introduction

Diarrheal disease is commonly encountered in clinical practice worldwide, causing substantial morbidity, mortality, and healthcare costs. These are also major public health problems due to their significantly increasing incidence over the last 25 years [1]. Most diarrhea diseases are associated with foodborne pathogens [2]. Approximately 800 foodborne disease outbreaks are reported in the United States each year, accounting for approximately 15,000 illnesses, 800 hospitalizations, and 20 deaths [3]. During the period from 2009 to 2015, among 2953 outbreaks with a single confirmed etiology reported in** the** USA, norovirus was the most common cause of outbreaks (1130; 38%) and outbreak-associated illnesses (27,623; 41%), followed by *Salmonella* with 896 outbreaks (30%) and 23,662 illnesses (35%). *Listeria*, *Salmonella*, and Shiga toxin-producing *Escherichia coli* (STEC) were responsible for 82% of all hospitalizations and 82% of deaths reported of the outbreaks in the period [4], whereas in the European Union for the year 2018, 26 member states reported a total of 5146 foodborne outbreaks, provoking 48,365 cases of illness, 4588 hospitalizations, and 40 deaths [5].

Although bacterial foodborne diseases are relatively well-documented, foodborne viruses may be the most under-recognized cause of outbreaks of gastroenteritis [6]. Norovirus is the most common agent involved in foodborne outbreaks causing gastroenteritis with high morbidity and mortality, mainly associated with developing countries [6]. In Mexico, as in many countries, the diseases caused by enteric viruses are not mandatory to be reported and are only informed along with other defined diseases causing diarrhea; consequently, data are underestimated [7]. 

Several parasites have also emerged as significant causes of foodborne and waterborne illness. The most common foodborne parasites in the European Union include *Cyclospora cayetanensis, Toxoplasma gondii,* and *Trichinella spiralis* [5], whereas in Latin America, a high incidence of amebiasis by *Entamoeba histolytica* is frequently reported [8].

Different clinical discomforts are associated with foodborne pathogens. Typically, diarrhea is the most common symptom; however, a wide range of other symptomatic problems can also be associated, such as vomiting; abdominal cramps; headaches; nausea; dry mouth and difficulty swallowing; and fluke-like symptoms, such as fever, chills, and backache [9]. In some cases, more severe problems are present, such as bloody diarrhea and inflammatory bowel disease (IBD). The incidence of IBD is rising in developing countries and is considered an emerging global disease [10]. In Mexico, the incidence of IBD has increased from 28.8 to 76.1 cases per 100,000 hospitalizations in the period from 1987 to 2006. Amoebiasis, giardiasis, and infections by *Salmonella, Shigella*, and *Campylobacter* are considered frequent causes of IBD in Mexico and Central America [10]. 

Nosocomial diarrhea is a common problem in hospitals [11]. Reports from the US during the 2009–2015 period indicated that from 5114 hospitalized patients suffering foodborne diseases with confirmed or suspected etiology, 82% were associated with *Salmonella*, Shiga-toxin *E. coli*, and *L. monocytogenes* [4]. However, variations in pathogens and proportions are found around the world; these vary depending on emerging pathogens, the food supply, healthy behaviors, and the number of people with heightened susceptibility to foodborne diseases [12]. Fincher et al. [13] determined that regional prevalence/absence of pathogens has a strong positive correlation with collective cultural practices, type of food consumed, behavior, and environment. 

The identification of these pathogens is necessary to design strategies for prompt treatment of the disease that they cause. In many developing countries, diarrheal diseases are one of the leading predictable causes of death [14]; however, to our knowledge, the cases and causes of diarrhea in hospitalized patients in Mexico are rarely documented, hence the present study.

## 2. Material and Methods

### 2.1. Sample Collection

A total of 240 feces samples were obtained from the third-level hospital José Eleuterio Gonzalez of the Universidad Autonoma de Nuevo Leon in Monterrey, Mexico. This hospital is a 450-bed tertiary care teaching hospital with an average hospital admittance rate of 26,500 patients annually. The study was carried out from January 2017 to June 2018. The scope of the study focused on hospitalized patients (suffering a diversity of diseases) without discrimination of age, gender, and diagnosis; however, all patients showed diarrhea at the time of admission (presence of diarrheal evacuations in accordance with the classification of Bristol type 5 to 7 was the only criteria of inclusion, other symptoms were not considered), with 3 or more evacuations in 24 h and after 72 h of admission. Stool samples were taken from patients before antimicrobial and antidiarrheal administration during the first three days of the disease. Samples were delivered to the laboratory in less than 1 h after collection.

The demographic, epidemiological, and clinical data of all patients were recorded. This study was approved by the Ethics and Research Committee of the Facultad de Medicina, Universidad Autonoma de Nuevo León in Monterrey, Mexico (approval no. PC18-00004). All patients granted their informed consent in writing or orally as approved by the Ethics Committee. In the case of minors, caretakers or guardians gave consent.

Macroscopic examination of feces was used to determine the Bristol scale, color, and presence or absence of mucus and/or blood. For this, an aliquot was taken from a representative section of the sample and was extended onto a slide, and a drop of saline solution was added. After mixing, a cover slide was applied to the sample and observed under the microscope. The presence of leukocytes and/or erythrocytes was recorded.

### 2.2. Microbial Detection

The presence of the following bacterial pathogens: *Clostridioides difficile*, *Shigella* spp., *Yersinia enterocolitica*, *Aeromonas* spp., and *Salmonella* [11], was determined by polymerase chain reaction (PCR). Norovirus was detected by quantitative PCR (qPCR) and parasites by microscopic observation and methods of concentration. 

#### 2.2.1. Strains Used as Controls

Strains of *C. jejuni, C. difficile* toxin B, *Shigella* spp., *Yersinia enterocolitica, Aeromonas* spp., and *Salmonella* spp. were used as controls. These strains were isolated from patients admitted to the hospital José Eleuterio Gonzalez of the Universidad Autonoma de Nuevo Leon in Monterrey, Mexico and identified as described in Section 2.2.2 and confirmed by Microflex MALDI-TOF MS system (Bruker Diagnostics, Ettingen, Germany). All strains were maintained in Brucella broth (Becton, Dickinson and Company, USA) at −80 °C. 

The culture was performed for the identification of bacteria as recommended by Lagier et al. [15]. In the case of norovirus, a G-block (double-stranded DNA fragments containing designed primer sequences) was devised containing the two sequences for the norovirus GI and GII (Table 1), using the genome NC_001959.2 and NC_029646.1 templates and Primer Blast, Amplify, and IDT.

#### 2.2.2. Bacterial Detection

DNA from the fecal samples was extracted using the commercial kit ISOLATE Fecal DNA (Bioline, London, UK) following the manufacturer′s instructions. In the case of control strains, a colony was taken and DNA was extracted using the commercial kit ISOLATE II Genomic DNA Kit (Bioline, London, UK). An aliquot of 3 μL of DNA from feces or control cultures was used as a template for PCR amplification. PCR was performed in a Biometra TOne Thermal Cycler (Analytic Jena AG, Montreal, QC H3C 0J7, Canada). Primers were used to amplify the 16S-23S rRNA (ITS) gene for *Salmonella* spp. [16], *vir*F gene for *Shigella* spp. [17], conserved regions of the 16S rDNA gene for *Campylobacter* [18], the *Yersinia* heat-stable enterotoxin (YST) gene [19], and conserved regions of 16S rDNA for *Aeromonas* [20] and the *C. difficile* toxin B (*tcd*B) gene [21], (Table 1).

For the PCR reaction, the total volume was 25 μL, consisting of 2.5 μL of 10 × buffer solution, 1.5 μL of MgCl_2_ (3 mM), 2 μL of dNTP mix (10 mmol/L), 2 μL each of forward and reverse primers (10 μmol/L), 0.2 μL of Taq polymerase (Bioline, London, UK), 12.3 μL of sterile water, and 2.5 μL of template DNA. The run conditions of each primer-set are described in Table 1. The amplification products were visualized using electrophoresis on a 1.5% agarose gel stained with 1 μg/mL ethidium bromide, compared with a standard (DNA Molecular Weight, Marker 100, Sigma Aldrich, St. Louis, MO, USA), and developed with a UV transilluminator (UVP Transilluminator M-20V, Analytik-Jena, Beverly, MA, USA). 

#### 2.2.3. Parasite Detection 

Stools were examined for the presence of blood or mucus for amoebas, or possible organisms, such as adult worms or proglottids of *Taenia* spp. One drop of stool was mixed with one drop of 5% formaldehyde solution (Sigma-Aldrich, St. Louis, MO, USA) to avoid deterioration of parasite structures. For microscopic examination, one drop of 0.85% saline solution and 1 drop of Lugol′s solution (Sigma Aldrich, Mexico City, Mexico) were placed on different points of a slide and mixed with a small sample of stools and homogenized. After placing a glass cover, these were observed under the microscope (Velab Microscopes, Pharr, TX, USA) at 40 × and 100 × objective. The presence of trophozoites, cysts of protozoa, eggs and/or helminth larvae was determined. Negative samples were subjected to fecal concentration techniques (flotation and sedimentation), followed by observation at the microscope according to Garcia et al. [23]. 

#### 2.2.4. Norovirus Detection 

Feces were diluted 1:10 with sterile phosphate-buffered saline (PBS), and after manual homogenization for 30 s, samples were centrifuged at 13,000 rpm for 5 min at 4 °C (Eppendorf centrifuge, mod. 5810R). An aliquot of supernatant (500 µL) was used for RNA extraction with TRIzol reagent as described by Heidary and Pahlevan-Kakhki [24]. The RNA concentration obtained was determined by measuring absorbance A_260_ using a Nanodrop system (mod. 2000, Thermo Scientific, Waltham, MA, USA), and cDNA was obtained by reverse transcription PCR (RT-PCR) using random primers following the manufacturer’s instructions.

Detection of norovirus GI and GII was evaluated by qPCR [22] using a 7500 fast thermocycler (Thermo Fisher Scientific Oy, Vantaa, Finland) and a TaqMan-based Probe (IDT, Coralville, IA, USA). The thermocycler conditions and primers are described in Table 1. 

### 2.3. Statistical Analyses

All of the assays were performed at least three independent times each in duplicate. *p*-values ≤0.05 were considered significant. Correlation assays were performed among bacteria, norovirus, and parasites by Fisher’s test and Cramer´s V correlation analysis using the statistical package SPSS (IBM SPSS Statistics 20, Armonk, NY, USA). Groups of age were analyzed against all variables (diseases, bacteria, parasites, and viruses).

## 3. Results

Stool samples from 240 patients were analyzed, 98 (40.8%) females and 142 (59.2%) males (Figure 1). They were arranged into seven groups depending on age: 0–10, 11–20, 21–30, 31–40, 41–50, 51–60, and 61–97. Although all patients suffered diarrhea, they were hospitalized due to a variety of illnesses, such as diarrhea, cancer, diabetes, and respiratory, heart, or kidney diseases (Figure 1). Of all the patients, in 40.4% of the samples (97/240), the presence of at least one of the pathogens analyzed was detected. However, the presence of norovirus was only analyzed in 182 samples (75.8%), so we cannot exclude the presence of norovirus in the rest of the samples. Furthermore, in 34.3% (49/143) of the samples in which no pathogen was detected, the primary diagnosis was problems related to the gastrointestinal tract.

From the total, 90 patients were diagnosed with gastrointestinal-related problems (with or without detection of pathogenic microorganism). Eighty-five (85/240, 35%) were diagnosed with diarrhea, and five (5/240, 2.1%) had CUCI (chronic ulcerative colitis idiopathic), a gastrointestinal-related disease (Figure 2). These patients belonged mainly (*p* < 0.05) to the age group 31–40 (16/29, 53%), followed by the age groups 21–30 (16/35, 47%), 0–10 (87/15, 46%), 11–20 (7/20, 35%), 41–50 (15/45, 33%), 61–97 (16/50, 32%), and 51–60 (12/46, 26%) (Figure 2).

From the pathogen-positive samples (*n* = 97), co-infections were detected in 13 (13.4%) patients, mainly parasite–norovirus or parasite–bacteria, and only one case of bacteria–bacteria co-infection was detected (*Salmonella-Aeromonas*). An important finding was that no co-infections were detected in the infant and young patients (age groups 0–10 and 11–20), but this was not true for older patients.

### 3.1. Bacteria

In this study, the presence of *Campylobacter* spp., *C. difficile*, *Shigella* spp., *Yersinia* spp., *Aeromonas* spp., and *Salmonella* spp. was analyzed by end-point PCR. From the total of samples analyzed, *Salmonella* was the pathogen most frequently found (10/240, 4.2%), followed by *C. difficile* (8/240, 3.33%), *Aeromonas* spp., (4/240, 1.6%), and *Shigella* spp. (3/240, 1.25%). The rest of the bacteria were detected in less than 1% of patients (Figure 3, Table 2).

Samples of patients of the age group 61–97 showed the highest frequency of *Salmonella* (4/50, 8%), with two of them diagnosed with diarrhea as the primary diagnostic, and the other two having diabetes and cancer (Figure 3). The samples from patients diagnosed with diarrhea showed *Salmonella* co-infections, one with *Aeromonas* spp., *C. difficile,* and norovirus GII, and the other with norovirus GII. The patient hospitalized with diabetes related-problems showed co-infection *Salmonella-Entamoeba coli.* No co-infections were detected in other age groups, (Figure 3).

*C. difficile* was the second more frequently bacterium found in the samples (8/240, 3.33%). The frequency distribution was relatively uniform (two positive samples in each of the age groups 21–30, 41–50, 51–60, and 61–97, Table 2). An interesting finding was that no patient from 0 to 20 years-old was positive for this bacterium. Furthermore, 5/8 patients presenting *C. difficile* were hospitalized due to problems not associated with gastrointestinal diseases, and those with gastrointestinal disease belonged to age groups 51–60 (1/8) and 60–97 (2/8) (Figure 3).

*Aeromonas* was found in four (1.6%) of the samples analyzed (Figure 3); the positive samples were distributed between age groups (one on each group): 11–20 (1/6, 16.6%), 31–40 (1/29, 3.44%), 51–60 (1/46, 2.17%) and 60–97 (1/50, 2%), however, since the number of patients of the age group 11–20 was lower (6), the percentage of positives of this bacterium was higher. Co-infections were detected in two of the four cases. Both in patients with gastrointestinal-related diagnosis, one belonging to age group 51–60 (diagnosed with peptic ulcer) who had co-infection with norovirus GII and one from age group 61–97 (diagnosed with diarrhea), showed co-infection with *Salmonella*. 

*Shigella* was only detected in 1.6% (3/240) of the total patients analyzed. All positive samples belong to patients of the age group 31–40 (3/29, 10.3%) (Figure 3). Two of them had diarrhea and other VIH. The sample from the VIH-positive patient showed co-infection with *Isospora* spp. 

*Campylobacter* spp. was detected in only 2 of the 240 patients (0.83%) and *Yersinia* spp. in only 1 sample (0.42%) (Figure 3). In all these samples, the cause of hospitalization (diabetes, and VIH) was not related to gastrointestinal problems.

### 3.2. Parasites

No helminths were detected in any of the samples analyzed. Detected protozoa included: *E. histolytica* (8/240, 3.33%), *E. coli* (8/240, 3.33%), *Blastocysts hominis* (2/240, 0.8%), *Cryptosporidium* (1/240, 0.4%), and *Isospora* (1/240, 0.4%) (Figure 4, Table 2). 

*E. histolytica* was commonly found in the 21–30 age group and older. This organism was found co-infecting with *Shigella* (1/240) and norovirus GII (3/240), whereas *E. coli* was detected in 6.66% (1/15) of patients in the 0–10 age group, 5% (1/20) in the 11–20 age group, 6.52% (3/46) in the 51–60 age group, and 4% (2/50) in the age group 65–97. *E. coli* was found co-infecting with *Salmonella* (1/240), *C. difficile* (1/240), and norovirus GII (2/240). 

*B. hominis* was detected in one patient of the age group 21–30 and another of the age group 51–60; the patient of the age group 21–30 was diagnosed with “diarrhea under study”, while the other patient was hospitalized due to diabetes complications. No co-infections of this parasite were detected. *Cryptosporidium* and *Isospora* were detected in one patient each; the first in the age group 21–30 and the second in the age group 31–40, both diagnosed with HIV. Both parasites were present as coinfections, *Cryptosporidium*–norovirus GII and *Isospora–Shigella*.

### 3.3. Virus: Norovirus

The presence of norovirus GI and GII was analyzed in 182/240 patients and were found in 53 (28.6%) of them (Figure 5, Table 2). GII was the most frequently found genogroup (49/182, 26.9%), whereas GI was detected in only 4/182 (2.2%) patients. 

Norovirus GI was detected only in females, one from the 0–10 age group, one from the 21–30 age group, and two from the 41–50 age group, and in all cases the patients were hospitalized with a diarrhea diagnosis. No co-infections with this genogroup were detected. 

No gender tendency for the positive samples of norovirus GII was detected (24 females vs. 25 males). The age group 0–10 exhibited the highest prevalence (9/15, 60%), followed by 51–60 (9/32, 28.1%), 41–50 (9/34, 26.9%), and 21–30 6/26, 23.1%); the rest of the age groups varied between 18.9 to 15.4% (Figure 5). From the norovirus-positive samples, 17/49 (34.7%) belong to patients whose primary diagnosis was related to gastrointestinal problems (diarrhea, CUCI, ulcerative disease, or colitis). Co-infections between norovirus GII and bacteria were present in only two cases (2/49, 4.1%), one case with *Aeromonas* and another with *C. difficile*. Parasite–norovirus GII co-infection was detected in 12.2% (6/49) of the samples analyzed; *E. histolytica* was the parasite most frequently found together with norovirus (3/49, 6.1%), followed by *E. coli* (2/49, 4.1%) and *Cryptosporidium* (1/49, 2.0%).

## 4. Discussion

A diversity of microorganisms causing diarrhea in hospitalized patients have been reported in various countries; as examples, in the Central Africa Republic, rotavirus, norovirus, astrovirus, *Shigella*/EIEC, and *Cryptosporidium hominis/parvum* were found as the most prevalent agents in 333 pediatric patients analyzed [25]. Reports from the US indicated that from 4731 hospitalized patients suffering foodborne diseases with confirmed or suspected etiology, 66.96% were associated with *Salmonella*, whereas 14.20% were affected with Shiga-toxin *E. coli*, 7.059% with *Listeria monocytogenes*, 2.83% with *Campylobacter,* and 2.28% with *Shigella* during the 2009–2015 period [4]. Slight differences were reported by Thomas et al. [26] from 11,600 hospitalized patients from Canada, where 66% of cases were due to bacteria, 31% to a virus, and 3% to parasites; from them, norovirus, nontyphoidal *Salmonella*, *Campylobacter,* and VTEC O157 accounted for 74% of all cases. Similar data were reported in France, where from 17,281 hospitalized patients suffering foodborne illness, 59% of cases were caused by bacteria, 38% by virus, and 3% by parasites; the pathogens more often found included *Campylobacter, Salmonella,* and norovirus, accounting for 76% of cases [27]. In Mexico, various pathogens are commonly reported as cause of diarrhea: *Salmonella*, *Shigella*, *Escherichia*, *Vibrio*, *Campylobacter*, *Yersinia, C. difficile, Aeromonas*, and several amoebas and viruses, such as norovirus [28,29].

The data obtained in this study show differences in the type and frequency of enteropathogens when compared to those reported. Norovirus (especially GII) was the leading cause (28.6%, 53/182) of hospitalized patients suffering from diarrhea due to either gastrointestinal-related or other health problems (which is higher than the average 17% previously reported from several studies [30]), followed by bacteria (11.7%) and parasites (8.3%). However, the percentage varies depending on the cause of hospitalization; for example, in those hospitalized for gastrointestinal-related problems, norovirus accounted for 70.1% (53/75), followed by *Salmonella* and *Aeromonas* (4.7%, 4/85 each), parasites (3.5%, 3/85), and *Shigella* and *C. difficile* (2.4%, 2/85 each). 

Although the Mexican government has little information about the incidence of norovirus in the country [28], in this study, norovirus was found at a high percentage in hospitalized patients and, in some cases, co-infecting with bacteria (or parasite). Since these conditions could worsen the clinical stage of patients, it is urgent to implement routine methods to detect this virus in food and people to reduce viral dispersion and prevalence in the population with special attention in immunocompromised patients.

Although in France *Campylobacter* leads in frequency [27] and Canada and USA are between the most often microorganism found in hospitalized patients due to gastrointestinal problems [4,26], in this study, this tendency was no detected, since *Campylobacter* and *Yersinia* was detected at very low frequency and only in hospitalized patients without gastrointestinal primary diagnoses.

These differences of pathogen prevalence can be due to many factors, such as the selection pressures imposed by pathogens influenced by the social behavior of many species, but also climatic and geographic aspects. Furthermore, worldwide variation in pathogen prevalence predicts cultural differences and practices [13].

In our study, co-infection was detected in 13 patients older than 21 years of age, and patients aged 0–20 years did not present co-infections. These results contrast with those reported by Breurec et al. [25], who found bacterial/viral co-infections in pediatric patients (mean age 12.9 months) in 10% of the samples from hospitalized patients in the Central Africa Republic. These differences could be associated with the prevailing conditions of poverty, health services, and nutrition levels in each country [25,28,29].

Acute and chronic infectious diarrhea affects more than 179 million individuals annually in the USA alone [2], ranking 4th among all causes of death worldwide in 2013 [28]. The Mexican Health System reported 2,000,188 consultations for gastrointestinal diseases in 2008, and data from 2003 indicated 4556 deaths caused by intestinal infections [29]. These diseases do not discriminate between age and social status, and the most vulnerable groups include children, the elderly, and immunosuppressed people [28]. Although there are several causes of diarrhea other than microbial infections, such as malnutrition and toxins found in contaminated food or water, intestinal infectious diseases are one of the most important diseases caused by a variety of pathogens, including protozoa, bacteria, helminths, and viruses. These pathogens are mostly transmitted via the fecal–oral route, through contaminated food and water, and occasionally from person to person [28,29]. 

In general, foodborne pathogens cause millions of cases of sporadic illness and chronic complications. The magnitude of this problem is demonstrated by the significant proportion of the 1.5 billion annual diarrheal episodes in children less than 3 years old that are caused by enteropathogenic microorganisms, which results in more than 3 million deaths per year [31]. However, it is estimated that the reported incidence of foodborne disease represents between 1 to 10% of the real incidence [32]. 

When people get sick, the economic impact can be very high; the FDA estimates that 1 to 3% of all cases of foodborne illness develop secondary diseases or complications that can be chronic and cause premature death [33]. The costs of these could be between USD 6.5 billion and USD 13.3 billion, regardless of the social implications when patients die or develop complications; thus, they never return to work, recover only part of their pre-illness productivity, or change to less demanding and poorly paid jobs [33].

The panorama may be more difficult for patients hospitalized for different conditions not related to gastrointestinal problems. Since they present diarrhea and pathogenic microorganisms, these can worsen the patient’s condition and make them more susceptible to aggravated conditions or complications due to their probable immunocompromised state. 

It is important to note that the data obtained in this study are specific to this hospital and we recognize that it would be difficult to make adequate comparisons with other regions of Mexico and with other countries. We have studied a small selection of a very diverse group of patients that included different ages and clinical conditions, but the type of patients, the hospital environment and the socioeconomic situation could differ in other regions of Mexico and other countries.

Due to the differences found between this study and data from other countries, it is important to make a more extensive and in-depth study of the agents that are most frequently found in this region and examine whether the data are similar in different regions of the country. These data will help researchers understand the behavior of microorganisms and can provide some guidelines to establish better management in this type of patient and to identify possible sources of contamination, such as foods, or dispersion mechanisms of the agents responsible for illness, to increase the efficiency of treatment or even prevent the infections. 

## Figures and Tables

**Figure 1 pathogens-09-00672-f001:**
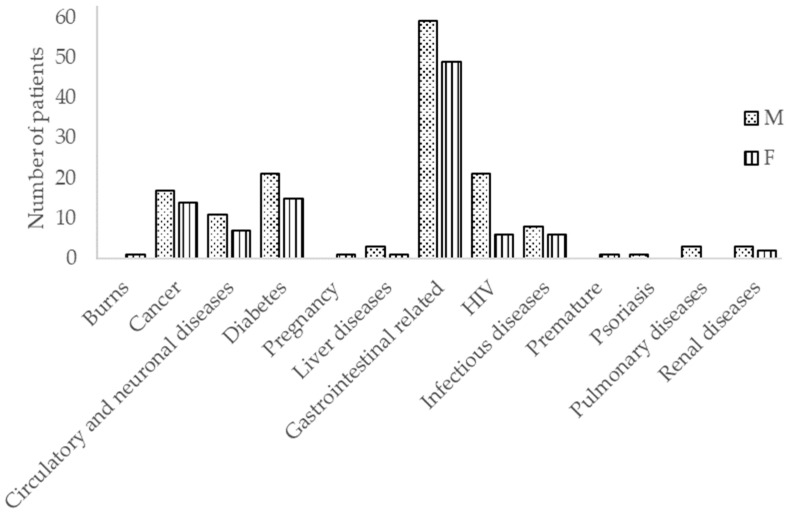
Primary diagnosis and gender of the patients studied. All patients showed diarrhea at the moment of this study.

**Figure 2 pathogens-09-00672-f002:**
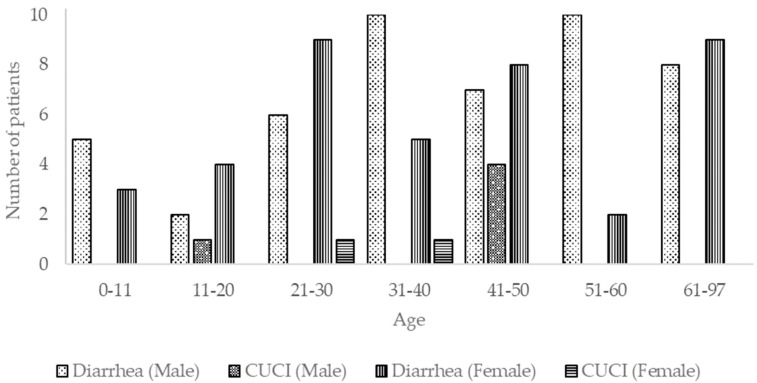
Frequency and gender of patients hospitalized with gastrointestinal-related problems included in this study.

**Figure 3 pathogens-09-00672-f003:**
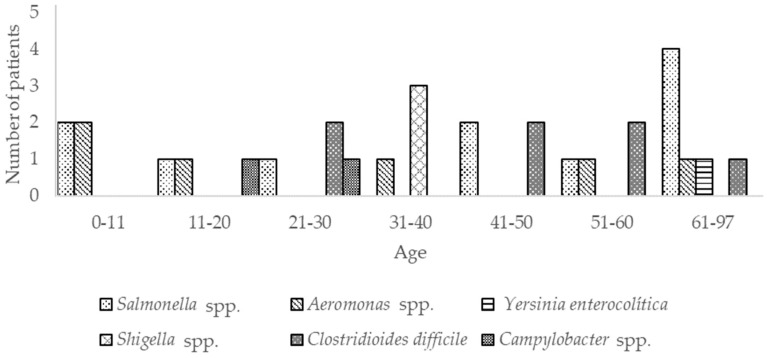
Frequency of pathogenic bacteria in feces of diarrheic-hospitalized patients.

**Figure 4 pathogens-09-00672-f004:**
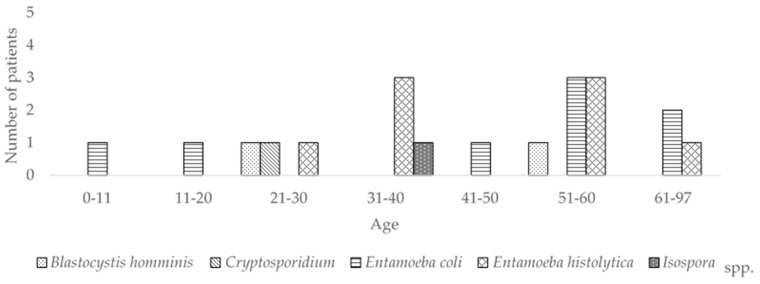
Frequency of parasites in feces of diarrheic-hospitalized patients.

**Figure 5 pathogens-09-00672-f005:**
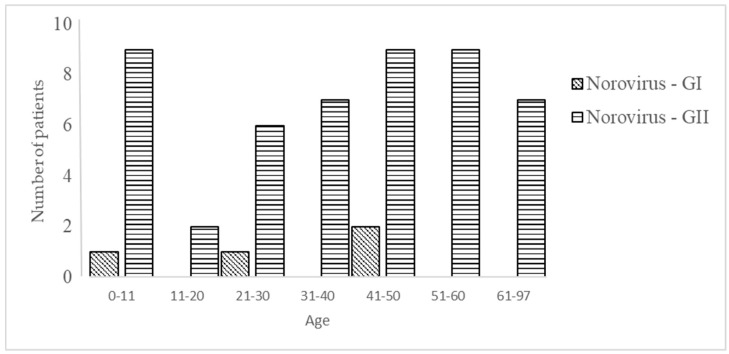
Number of patients with norovirus GI and GII in diarrheic-hospitalized patients.

**Table 1 pathogens-09-00672-t001:** Primers used and PCR conditions for analysis of bacteria and norovirus in fecal samples of hospitalized patients.

Microorganism	Target Gene	Primers Sequence	Amplicon Size (bp)	Run Conditions	Ref.
*Salmonella* spp.	ITS	5′-TATAGCCCCATCGTGTAGTCAGAAC-3′ F	312	5 min at 94 °C, 35 cycles of 30 s at 94 °C, 35 s at 61 °C, and 35 s at 72 °C, and final extension at 72 °C for 5 min.	[16]
		5′-TGCGGCTGGATCACCTCCTT-3′ R			
*Shigella* spp.	*Vir*F	5′-AGCTCAGGCAATGAAACTTTGAC-3′ F	628	5 min at 95 °C. 30 cycles of 30 s at 95 °C, 30 s at 58 °C, and 30 s at 72 °C. and final extension at 72 °C for 5 min.	[17]
		5′-TGGGCTTGATATTCCGATAAGTC-3′ R			
*Campylobacter* spp.	16S rRNA	5′-CACGTGTCACAATGGCATAT-3′ F	115	6 min at 95 °C, 30 cycles of 30 s at 95 °C, 30 s at 59 °C, and 30 s at 72 °C, and final extension at 72 °C for 7 min.	[18]
		5′-GGCTTCATGCTCTCGAGTT-3′ R			
*Y. enterocolitica*	YST	5′-GTTAATGCTGTCTTCATTTGGAGC-3′ F	145	2 min at 94 °C, 40 cycles of 30 s at 92 °C, 30 s at 59 °C, and 30 s at 72 °C, and final extension at 72 °C for 5 min.	[19]
		5′-GACATCCCAATCACTACTGACTTC-3′ R			
*Aeromonas* spp.	16S rRNA	5′-CTACTTTTGCCGGCGAGCGG-3′ F	935	5 min at 95 °C; then 35 cycles of 1 min at 94 °C, 30 s at 68 °C, and 45 s at 72 °C, and final extension at 72 °C for 5 min.	[20]
		5′-TGATTCCCGAAGGCACTCCC-3′ R			
*C. difficile*toxin B	*tcd*B	5′-GGAAAAGAGAATGGTTTTATTAA-3′ F	135	3 min at 95 °C, 10 cycles of 30 s at 95 °C, 30 s at 65 °C, and 30 s at 72 °C, and final extension at 95 °C for 30 s.	[21]
		5′-ATCTTTAGTTATAACTTTGACATCTTT-3′ R			
norovirus GI		5′-GCCATGTTCCGITGGATG-3′ F	96	10 min at 95 ºC, 40 cycles of 15 s at 95 ºC and 1 min at 60 ºC.	[22]
		5′-TCCTTAGACGCCATCATCAT-3′ R			
		FAM-5′-TGTGGACAGGAGATCGCAATCC-3′			
Norovirus GII		5′-CAAGAGTCAATGTTTAGGTGGATGAG-3′ F	89	10 min at 95 ºC, 40 cycles of 15 s at 95 ºC and 1 min at 60 ºC.	[22]
		5′-TCGACGCCATCTTCATTCACA-3′ R			
		Cy5-5′-TGGGAGGGCGATCGCAATCT-3′			

**Table 2 pathogens-09-00672-t002:** Number of patients harboring pathogenic microorganisms by age grouping.

Pathogen	Number of Patients (*n* = 240)						
	Age Group						
	0–10 (*n* = 15)	11–20 (*n* = 20)	21–30 (*n* = 35)	31–40 (*n* = 29)	41–50 (*n* = 45)	51–60 (*n* = 46)	61–97 (*n* = 50)
**Bacteria**							
*Campylobacter* spp.	ND	1	1	ND	ND	ND	ND
*Clostridioides* *difficile*	ND	ND	2	ND	2	2	1
*Shigella* spp.	ND	ND	ND	3	ND	ND	ND
*Yersinia enterocolitica*	ND	ND	ND	ND	ND	ND	1
*Aeromonas* spp.	2	1	ND	1	ND	1	1
*Salmonella* spp	2	1	1	ND	2	1	4
**Parasites**							
*Entamoeba histolytica*	ND	ND	1	3	ND	3	1
*Entamoeba coli*	1	1	ND	ND	1	3	2
*Blastocysts hominis*	ND	ND	1	ND	ND	1	ND
*Cryptosporidium* spp.	ND	ND	ND	1	ND	ND	ND
*Isospora* spp.	ND	ND	ND	1	ND	ND	ND
**Virus**							
norovirus GI	1	ND	1	ND	2	ND	ND
norovirus GII	9	2	6	7	9	9	7

ND: Non detected.

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
