# Peer review of "Norovirus Is the Most Frequent Cause of Diarrhea in Hospitalized Patients in Monterrey, Mexico"

_pathogens, 2020, doi:10.3390/pathogens9090672_

Round 1
Reviewer 1 Report
In this manuscript the authors address an important issue of infections and co-infections that cause diarrhea in hospitalized patients. This has a good structure, is well written and easy to follow. The authors provide relevant information on the topic and emphasize the lack of detailed reports on infections with certain pathogens which leads to underestimation of cases and, thus, a poor comprehension of the risks in a clinical setting. The authors test faecal samples from hospitalized patients with diarrhea and test for presence of specific pathogenic agents (virus, bacteria, parasite). This work reveals a novel pattern of infection in hospitalized patients, regarding both type of pathogen and frequency, which differs from reports from other countries and continents. The authors describe that approximately 40% of the sample population tested positive to at least one of the pathogens, with higher incidence of Norovirus, followed by bacteria and parasites. The study reveals that co-infection were exclusive to patients older than 20 years of age. This study also reveals how the incidence of a pathogenic agent such as Norovirus is very poorly in Mexico (study location) but is indeed very prevalent in the sample population analysed here. These data reinforce the need to evaluate the risk of nosocomial infections and improve the treatment approach for hospitalized patients.
Specific comments:
- In the “Methods” the authors describe the criteria for patient selection, but one thing is not clear to me. For the patients that were admitted in the hospital without an intestinal disorder, did they develop diarrhea? Were data collected in respect to a patient with an unrelated disorder coming into the hospital infected already? Or were all the positive samples confirmed nosocomial infections?
- In one paragraph of the “Discussion” the authors describe their data on co-infection in younger patients, which contrasts with data from another study. However, not much is added regarding whether that is the only study available, or discussing the origin of such differences.
- There is some heterogeneity in text font/size/color, please correct this (e.g. L39, L83, etc).
- L 43: “whereas in the European Union for the year 2015, 26 member states reported a total of 4,362 foodborne outbreaks, provoking 45,874 cases of illness, 3,892 hospitalizations, and 17 deaths [5].”
This section could include updates from more recent data from the annual reports of the European Food Safety Authority (https://efsa.onlinelibrary.wiley.com) such as: https://efsa.onlinelibrary.wiley.com/doi/epdf/10.2903/j.efsa.2019.5926
- In the “Materials and Methods” while the adequate information is present, I believe 2.3, 2.4, 2.5 and 2.6 should be sub-sections of 2.2 (2.2.1, 2.2.2, etc) since they are extensions of the entire microbial detection procedure, and is easier for the reader to follow. Also, some readers might find the use of the described strains in 2.3 as controls vague. More detail here would be useful.
- L 99: “The presence of bacterial pathogens, including Clostridioides difficile, Shigella spp., Yersinia enterocolitica, Aeromonas spp., and Salmonella [11], was determined by polymerase chain reaction (PCR).”
- The authors here mention the list of pathogens tested “includes” the following. Does this mean that there were others tested, but only with negative results? If not, what was the criteria for pathogen selection? In other parts of the manuscript the authors describe other pathogens usually prevalent in patients with diarrhea including in Mexico (Escherichia, Vibrio, Campylobacter, Listeria, etc.). Where none of these tested?
- Figure 2: The legend is imperceptible unless reader checks Figure 1 legend and assume same order.
- Figure 3/4/5: I personally find these types of legend very hard to tell apart. Better bar pattern or bigger legend icons would help the reader interpret the data.
- L 285: “NoV was found at a high percentage in hospitalized patients and, in some cases, co-infecting with bacteria (or parasite), worsening the clinical stage of patients.”
- Is this observation speculation, based on literature, or were there any data collected regarding the medical state of the patients?
Author Response
1.- In the “Methods” the authors describe the criteria for patient selection, but one thing is not clear to me. For the patients that were admitted in the hospital without an intestinal disorder, did they develop diarrhea? Were data collected in respect to a patient with an unrelated disorder coming into the hospital infected already? Or were all the positive samples confirmed nosocomial infections? R= Thank you for your observation. The patients when they entered the hospital suffered from a variety of illnesses, yet they all had diarrhea. Data of non-diarrheal disorders were documented and are described in the MS. We are clarifying this information in the text.
2.- In one paragraph of the “Discussion” the authors describe their data on co-infection in younger patients, which contrasts with data from another study. However, not much is added regarding whether that is the only study available, or discussing the origin of such differences. R= We have updated the paragraph and included more discussion. Unfortunately, there is little information in the literature on coinfections in hospitalized patients.
3.- There is some heterogeneity in text font/size/color, please correct this (e.g. L39, L83, etc). R= We are sorry for these typographical errors. The font has been homogenized throughout the text.
4.- L 43: “whereas in the European Union for the year 2015, 26 member states reported a total of 4,362 foodborne outbreaks, provoking 45,874 cases of illness, 3,892 hospitalizations, and 17 deaths [5].”
This section could include updates from more recent data from the annual reports of the European Food Safety Authority (https://efsa.onlinelibrary.wiley.com) such as: https://efsa.onlinelibrary.wiley.com/doi/epdf/10.2903/j.efsa.2019.5926
R= Data has been updated. Reference was also changed
5.- In the “Materials and Methods” while the adequate information is present, I believe 2.3, 2.4, 2.5 and 2.6 should be sub-sections of 2.2 (2.2.1, 2.2.2, etc) since they are extensions of the entire microbial detection procedure, and is easier for the reader to follow. Also, some readers might find the use of the described strains in 2.3 as controls vague. More detail here would be useful. R= The sections were reorganized accordingly, and more details about isolation of the control strains were included: These strains were isolated from patients admitted to the hospital José Eleuterio Gonzalez of the Universidad Autonoma de Nuevo Leon in Monterrey, Mexico and identified by biochemical tests and PCR and confirmed by Microflex MALDI-TOF MS system (Bruker Diagnostics, Germany).
6.- L 99: “The presence of bacterial pathogens, including Clostridioides difficile, Shigella spp., Yersinia enterocolitica, Aeromonas spp., and Salmonella [11], was determined by polymerase chain reaction (PCR).” The authors here mention the list of pathogens tested “includes” the following. Does this mean that there were others tested, but only with negative results? If not, what was the criteria for pathogen selection? In other parts of the manuscript the authors describe other pathogens usually prevalent in patients with diarrhea including in Mexico (Escherichia, Vibrio, Campylobacter, Listeria, etc.). Where none of these tested? R= Although many bacteria have been responsible for gastrointestinal problems, the pathogenic bacteria determined in the study corresponded to the most reported in the hospital environment. The text was modified to clarify this information.
7.- Figure 2: The legend is imperceptible unless reader checks Figure 1 legend and assume same order. R= Figures were modified and improved.
8.- Figure 3/4/5: I personally find these types of legend very hard to tell apart. Better bar pattern or bigger legend icons would help the reader interpret the data. R= The figures were rearranged to make it more understandable and the scales were increased to more clearly visualize the number of patients. Font type was also homogenized
9.- L 285: “NoV was found at a high percentage in hospitalized patients and, in some cases, co-infecting with bacteria (or parasite), worsening the clinical stage of patients.” Is this observation speculation, based on literature, or were there any data collected regarding the medical state of the patients? R= We have modified this paragraph to provide clear information. “Although the Mexican government has little information about the incidence of NoV in the country [27], in this study, NoV was found at a high percentage in hospitalized patients and, in some cases, co-infecting with bacteria (or parasite). Since these conditions could worsen the clinical stage of patients, it is urgent to implement routine methods to detect this virus in food and people to reduce viral dispersion and prevalence in the population with special attention in immunocompromised patients”.
Reviewer 2 Report
The paper is intresting and well written, I have a consideration, why the authors did not cosider the copresence of two or more bactial, virus or parasites in the same sample?
I have few suggestion:
Material and methods section:
lines 83-85: these lines are not in Time new Roman
section 2.2 and 2.4 can be integrated
section 2.6 line 153: termocycler conditions and primers are descridebed in Table 1.
Results section:
Figure 1: reduce the scale of "number of patient" to avoid losing the number of patients between 0 and 10.
Figure 2: reduce the scale of "number of patient" to avoid losing the number of patients between 0 and 5.
Figure 3-4 and 5: it's would be better use the same graphic conditions.
Author Response
1.- The paper is intresting and well written, I have a consideration, why the authors did not consider the copresence of two or more bactial, virus or parasites in the same sample? R= Thanks for the comment. Actually we analyzed the presence of all studied pathogens in every sample; in the MS was used the term coinfection to express it.
2.- lines 83-85: these lines are not in Time new Roman. R= It was modified
3.- section 2.2 and 2.4 can be integrated. R= These sections were arranged as subgroups of the microbial detection section
4.- section 2.6 line 153: termocycler conditions and primers are descridebed in Table 1. R= The worlds “and primers” were added in the text
5.- Figure 1: reduce the scale of "number of patient" to avoid losing the number of patients between 0 and 10. R= Figures were rearranged to make it more understandable and the scales were increased to more clearly visualize the number of patients.
6.- Figure 2: reduce the scale of "number of patient" to avoid losing the number of patients between 0 and 5. R= The figures were rearranged to make it more understandable and the scales were increased to more clearly visualize the number of patients. Font type was also homogenized
7.- Figure 3-4 and 5: it's would be better use the same graphic conditions. R= The figures were rearranged, and the scales were increased to more clearly visualize the number of patients. Font type was also homogenized
Reviewer 3 Report
Casillas-Vega et al describe the microbiology test results of 240 stools from patients with diarrhea. It would be nice to include the age-range in the abstract. Norovirus should not be capitalized and not abbreviated as ‘NoV’ as it is a single word: just write ‘norovirus’
With norovirus detected in 28.6% of 182 samples (of the 240 samples) the title of this paper should be as follows: “Norovirus is the most frequent cause of diarrhea in hospitalized patients in Monterey, Mexico”
Include ‘in Mexico’ in line 75
Lines 83 and 84: ‘evacuations’ Are the authors referring to the number of episodes of diarrhea? Was vomiting excluded? Was only acute gastroenteritis included or also chronic diarrhea?
Paragraph 3.2 Microbial detection. Please insert the reference for the norovirus real-time RT-PCR [32]
Line 109. What is a G-block? Please clarify.
3.3 Virus: NoV replace with 3.3 Norovirus
Lines 323-333: this paragraph can be removed. The data in this study are clearly supporting the importance of norovirus as the cause of diarrhea in hospitalized patients with diarrhea and the authors should highlight that testing for this virus should become routine in Mexican hospitals.
I highly recommend to include a review paper on the global burden of norovirus, for example,
Ahmed SM, Hall AJ, Robinson AE, et al. Global prevalence of norovirus in cases of gastroenteritis: a systematic review and meta-analysis. Lancet Infect Dis. 2014;14(8):725-730.
